# Determination of Antimicrobial Resistance Patterns in Salmonella from Commercial Poultry as Influenced by Microbiological Culture and Antimicrobial Susceptibility Testing Methods

**DOI:** 10.3390/microorganisms9061319

**Published:** 2021-06-17

**Authors:** Xi Wang, W. Evan Chaney, Hilary O. Pavlidis, James P. McGinnis, J. Allen Byrd, Yuhua Z. Farnell, Timothy J. Johnson, Audrey P. McElroy, Morgan B. Farnell

**Affiliations:** 1Department of Poultry Science, Texas A&M AgriLife Research, College Station, TX 77843, USA; wangxi@swun.edu.cn (X.W.); yfarnell@tamu.edu (Y.Z.F.); amcelroy@tamu.edu (A.P.M.); 2Diamond V, Cedar Rapids, IA 52404, USA; echaney@diamondv.com (W.E.C.); hpavlidis@diamondv.com (H.O.P.); jmcginnis@diamondv.com (J.P.M.); Allen.Byrd2@usda.gov (J.A.B.); Tim_Johnson@diamondv.com (T.J.J.)

**Keywords:** antimicrobial resistance, antimicrobial susceptibility testing, poultry, *Salmonella*

## Abstract

Monitoring antimicrobial resistance of foodborne pathogens in poultry is critical for food safety. We aimed to compare antimicrobial resistance phenotypes in *Salmonella* isolated from poultry samples as influenced by isolation and antimicrobial susceptibility testing methods. *Salmonella* isolates were cultured from a convenience sample of commercial broiler ceca with and without selective broth enrichment, and resistance phenotypes were determined for 14 antimicrobials using the Sensititre^®^ platform and a qualitative broth breakpoint assay. The broth breakpoint method reported higher resistance to chloramphenicol, sulfisoxazole, and the combination of trimethoprim and sulfamethoxazole, and lower resistance to streptomycin as compared to the Sensititre^®^ assay in trial one. Selective enrichment of samples containing *Salmonella* in Rappaport-Vassiliadis broth reported lowered detectable resistance to amoxicillin/clavulanic acid, ampicillin, azithromycin, cefoxitin, ceftriaxone, nalidixic acid, and meropenem, and increased resistance to streptomycin and tetracycline than direct-plating samples in trial one. Using matched isolates in trial two, the Sensititre^®^ assay reported higher resistance to chloramphenicol and gentamicin, and lower resistance to nalidixic acid as compared to the broth breakpoint method. These results suggest methodology is a critical consideration in the detection and surveillance of antimicrobial resistance phenotypes in *Salmonella* isolates from poultry samples and could affect the accuracy of population or industry surveillance insights and intervention strategies.

## 1. Introduction

*Salmonella* is one of the most common foodborne pathogens. The intestinal tract of poultry and other food animals is considered the main foodborne *Salmonella* reservoir [1,2]. An increasing incidence of antimicrobial resistance (AMR) has been reported in poultry *Salmonella* isolates where antibiotics are extensively used in production systems [3,4]. Although the link between antimicrobial usage in food animals and clinical treatment failures in human salmonellosis cases remains controversial and inconclusive, the United States Food and Drug Administration (FDA) has fostered antimicrobial stewardship practices to reduce overall resistance since 2018, including ending the use of medically important drugs as growth promoters and requiring a veterinary prescription for medically important drugs [5]. Nevertheless, poultry *Salmonella* isolates carrying AMR genes have the potential to pass to consumers along the farm-to-fork continuum [6]. Those resistant *Salmonella* isolates could threaten public health when their resistance phenotypes interfere with drug treatments or when they transmit resistance determinants to other pathogens [7]. Monitoring the prevalence and the evolution of foodborne pathogen AMR in poultry and other animals has a critical impact on food safety and public health.

The interagency National Antimicrobial Resistance Monitoring System for enteric bacteria (NARMS) was established in 1996 to track AMR in foodborne pathogens including *Salmonella* isolated from live production, harvest, and retail products [8]. The NARMS surveillance program tracks *Salmonella* isolates to determine antibiotic susceptibility as well as resistance genotypes by pulsed-field gel electrophoresis and whole-genome sequencing [9]. A study of 200 *Salmonella enterica* isolates indicated the correlation of genotypic and phenotypic AMR was 87.61% and 97.13% for sensitivity and specificity, respectively [10]. Genotypic-based methods have a promise for rapid detection; however, phenotypic methods have an advantage of accuracy when the resistance is caused by multiple mechanisms [11].

Common antimicrobial susceptibility tests include the agar dilution method, broth dilution methods, gradient diffusion method (on a Mueller-Hinton agar plate), disk diffusion test, and automated instrumentation platforms [12]. The Sensititre^®^ system (TREK Diagnostic Systems, Cleveland, OH, USA) is an automated and standardized method that has been developed and adapted into the NARMS program for surveillance [12]. One isolate, from a positive sample, is selected, prepared, and evaluated for the minimum inhibitory concentration (MIC) level across the entire NARMS panel of antimicrobial compounds per each Sensititre^®^ plate and resistance or susceptibility qualitatively called at the included breakpoint concentration [13]. Usually, a single isolate from a positive sample is submitted and tested through the Sensititre^®^ platform due to expense. Feye and colleagues reported a qualitative broth breakpoint assay to conduct antimicrobial susceptibility testing (AST) of a large number of isolates rapidly and inexpensively [14]. Up to 96 *Salmonella* isolates, or more depending on plate configurations, can be inoculated into each well of a 96-well plate prefilled with broth and the tested antimicrobial at the Clinical and Laboratory Standards Institute (CLSI)/NARMS recommended resistance breakpoint concentrations [14]. After 18 to 24 h of incubation, the growth of the inoculated colony could be determined to indicate resistance or susceptibility. While the broth breakpoint method does not produce MIC data, both methods can be compared to estimate resistance prevalence using the current breakpoint concentrations.

We proposed that this broth breakpoint method, facilitating more isolates to be tested, was a more representative means of evaluating phenotypic AMR patterns from a food animal population where foodborne pathogens, such as *Salmonella*, are expected to be found in much higher prevalence and concentration with greater diversity. Therefore, we conducted two trials to compare phenotypic AMR observations in *Salmonella* isolated from within a broiler population at harvest as influenced by the culture and AST methodology deployed.

## 2. Materials and Methods

### 2.1. Study Design

Two trials were conducted via a similar workflow (Figure 1). All experimental procedures were reviewed and approved by the Texas A&M University Institutional Biosafety Committee (IBC number: 2019-073). A convenience sample of ceca was collected from market-aged broilers at harvest provided by a single poultry integrator in the Southeastern United States on 8 December 2018 (trial one) and 27 August 2019 (trial two). Samples were collected in the processing plant and shipped at 2–8 °C overnight to the laboratory. Upon receipt, samples were prepared and screened by commercial PCR for the presumptive detection of *Salmonella enterica* (BAX^®^ PCR Assay, Hygiena LLC., Camarillo, CA, USA). Sample retains of presumptive positive samples were then processed through serial dilution and direct-plating onto *Salmonella* selective agar (brilliant green agar (BGA) with 25 µg mL^−1^ novobiocin in trial one and xylose lysine deoxycholate agar (XLD) in trial two), isolation by secondary selective enrichment in Rappaport-Vassiliadis (RV) broth and streak plating to the selective agar plates. *Salmonella* isolates harvested via direct-plating and enrichment methods were assayed for antimicrobial susceptibility against the NARMS panel of antimicrobials using both the Sensititre^®^ automated system and the broth breakpoint method. The broth assessment was conducted only at the CLSI/NARMS breakpoint concentration of each evaluated antimicrobial compound.

In trial one, 57 presumptively positive samples and 3 indeterminate samples were obtained from 206 broiler ceca through PCR screening (*Salmonella* 2 PCR Assay, KIT2012, Hygiena LLC). These 60 samples were processed with direct plating, and selective enrichment and isolation. From the pool of 60 positive samples, we collected *Salmonella* isolates from 20 direct plating plates and 38 RV enrichment plates. Up to 94 *Salmonella* isolates per each of the 20 direct-plating positive samples were tested using the broth breakpoint assay (Figure 1, left). Three isolates from each RV enrichment were also assessed via the breakpoint assay. Another three isolates from the same direct-plating plate and three isolates from the RV enrichment streak plate were processed to perform the Sensititre^®^ method.

For trial two, matched *Salmonella* isolates were assessed by AST to eliminate the possible effects of isolate sample size and isolate diversity. The interaction and major effects of the culture method and AST method on the phenotypic AMR prevalence was evaluated via a 2 × 2 factorial setting of treatments (Figure 1, right). A total of 15 *Salmonella* positive samples were selected through a PCR screening (*Salmonella* Real-time PCR Assay, KIT2006, Hygiena LLC), direct-plating, and RV selective enrichment (*n* = 15). A total of 16 isolates (8 from direct plating and 8 from selective enrichment) were randomly selected per cecum sample and each individually regrown and evaluated for susceptibility via Sensititre^®^ and the broth breakpoint assays in a paired manner.

### 2.2. Sample Preparation

Cecum samples were collected and packed individually in an enclosed sterile Whirl-Pak^®^ bag (Nasco, Fort Atkinson, WI, USA) and transported on ice packs at 2–8 °C. Samples were prepared as a slurry by lacerating the entire cecum and massaging the contents in 10 mL of buffered peptone water (BPW, BD Difco^TM^, Franklin Lakes, NJ, USA) for 30 s using a Stomacher^®^ blender (Seward, Bohemia, NY, USA). One mL of the ceca slurry was diluted with nine mL of BPW to prepare a 1:10 dilution which was then incubated and screened by PCR for *Salmonella* detection. The remaining slurry was stored at 4 ± 1 °C until further analysis.

### 2.3. PCR Screening

Rapid screening by PCR (BAX^®^ System, Hygiena LLC) was utilized to identify cecum slurries presumptively positive for *Salmonella*. Briefly, the 1:10 dilution of the slurry was incubated at 37 °C for 18 h to pre-enrich bacteria. Bacterial genome fragments were exposed by lysing bacterial cells in 5 µL of pre-enriched suspension. The *Salmonella* target sequence was amplified by the PCR procedure on a Q7 instrument (Hygiena LLC). Trial one utilized an endpoint procedure (BAX^®^ System *Salmonella* 2, KIT2011, Hygiena LLC) and trial two utilized a real-time detection procedure (BAX^®^ System Real-Time *Salmonella* Assay, KIT2006, Hygiena LLC). Presumptive presence (positive) or absence (negative) of *Salmonella* was reported based on the fluorescent signal and software associated with each assay.

### 2.4. Salmonella Direct Plating and Enrichment

Samples presumptively positive by PCR were then cultured for *Salmonella* in parallel by direct plating of non-enriched sample dilutions and by selective secondary broth enrichment with streak plating for isolation. Direct plating was conducted by a ten-fold serial dilution of the primary cecal slurry using BPW spread plating dilutions onto BGA (trial one) or XLD agar plates (trial two). Selective secondary enrichment and isolation were performed by transferring 100 µL of the PCR positive BPW pre-enrichment to 10 mL of RV broth (Criterion^TM^, Hardy Diagnostics, Santa Maria, CA, USA) which was incubated at 42 ± 1 °C for 24 h. Enriched RV broth was then three-way struck onto selective agar plates for isolation via a 10 µL disposable sterile loop (Hach, Loveland, CO, USA). Cecal slurry, dilutions, and enrichment were plated onto BGA (BD Difco^TM^) with 25 µg mL^−1^ novobiocin in trial one and XLD (BD Difco^TM^) agar in trial two. The BGA plated samples were incubated at 37 ± 1 °C for 48 h and the XLD plates for 24 h followed by room temperature incubation for full-color development. Morphologically typical *Salmonella* colonies represented as white to red colonies surrounded by red zones on BGA plates and red colonies with black centers on XLD plates were confirmed biochemically (triple sugar iron agar and lysine iron agar, BD Difco^TM^) and serologically (*Salmonella* O antiserum, Poly A-I and Vi, BD Diagnostic, Spark, MD, USA).

### 2.5. Broth Breakpoint Assay

The broth breakpoint assay was modified from a resistance breakpoint assay described by Feye et al. [14]. Briefly, a sterile 96-well plate (VWR^®^ tissue culture plate, Avantor^®^, Wayne, PA, USA) was filled with 200 µL of Mueller-Hinton broth (BD Difco^TM^) in each well. A total of 94 colonies along with 2 controls (sterile blank and one *Salmonella* positive control, wild strain) were individually inoculated into the last two wells by sterilized toothpicks. After overnight incubation at 37 ± 1°C, the OD_600_ absorbance reading (Synergy H1 plate reader, BioTek Instruments Inc., Winooski, VT, USA) of the plate reached over 0.3, which corresponded to 3 × 10^8^ cfu of propagated *Salmonella* mL^−1^ in each well. The OD_600_ absorbance reading in the sterile blank was 0.046 ± 0.001, which indicated no bacterial growth. Antimicrobials were prepared at the CLSI/NARMS resistance breakpoint concentrations (NARMS panel, Table 1) in Mueller-Hinton broth. The tested antimicrobial solution was aliquoted into 200 µL in each well of another sterile 96-well plate for each compound. Propagated *Salmonella* isolates were inoculated into each antimicrobial plate utilizing a 96 well pin replicator tool (Boekel Scientific^TM^, Feasterville, PA, USA). The replicator was rinsed, flame sterilized, and cooled before each replicate inoculation. Plates were covered and incubated at 37 ± 1 °C for 24 h and the OD_600_ reading was recorded. When the absorbance reading was larger than 0.10, the bacterial growth in the antimicrobial plate was visible and considered resistant at the respective breakpoint.

### 2.6. Automated AST System

The Sensititre^®^ platform was utilized following the manufacturer’s instructions. Antimicrobial susceptibility testing of *Salmonella* was determined using the CMV4AGNF susceptibility plate (Thermo Fisher Scientific, Waltham, MA, USA). Briefly, an individual isolate was randomly chosen from a direct-plating or enrichment, struck onto a blood agar plate (Trypticase^TM^ soy agar with 5% sheep blood, BD BBL^TM^, Sparks, MD), and incubated at 35 ± 1 °C for 24 h. To achieve the optimal concentration, one or two pure colonies were suspended into five mL of deionized water and adjusted to a 0.5 McFarland standard equivalent using a Sensititre^®^ nephelometer (Thermo Scientific^TM^), and 10 µL of the suspension was transferred to 11 mL of Mueller-Hinton broth (Thermo Scientific^TM^). Fifty µL of the broth was inoculated into each well of the Sensititre^®^ plate by an automated inoculation delivery system (Sensititre AIM^TM^, Thermo Scientific^TM^) and incubated at 37 ± 1 °C for 24 h. Sensititre^®^ plates were read automatically (OptiRead^TM^ Automated Fluorometric Plate Reading System, Thermo Scientific^TM^) and confirmed manually (Vizion^TM^ Digital MIC Viewing System, Thermo Scientific^TM^). The growth of the colony in the tested antimicrobial solution at the breakpoint resistance concentration was determined for resistance.

### 2.7. Data Analysis

In trial one, AMR prevalence (% resistant isolates) outcomes observed by the two AST methods were compared by Chi-square analysis using the SAS FREQ procedure in trial one [15]. To reduce the possible effects of sampling size on AMR, AMR prevalence outcomes of the equal number of *Salmonella* isolates were randomly selected from the broth breakpoint assay and then compared to the Sensititre^TM^ assay. Randomization was conducted by a random number producer in Excel (Microsoft, Redmond, WA, USA). In trial one, 14 cecum slurry samples were positive for *Salmonella* colonies by direct plating and enrichment. Prevalence of resistance generated from the Sensititre^®^ method was compared for differences between direct plating and enrichment using a nonparametric method, the Wilcoxon signed-rank test in SAS NPAR1WAY procedure. In trial two, the major effects of susceptibility testing methodology and culturing methodology and their interactions on poultry *Salmonella* AMR were evaluated. A two-way ANOVA analysis was conducted by the SAS GLM procedure. All significance levels were set at α = 0.05.

## 3. Results

In trial one, 206 broiler cecum samples were screened by PCR assay and presumptive samples were cultured in parallel by direct plating and selective enrichment methods. Fifty-seven samples were PCR positive and three were indeterminate. Two of the three indeterminate samples were confirmed to be positive through cultural isolation, which indicated *Salmonella* prevalence in the convenience sample broiler population was approximately 28.6% (59/206, either PCR or culture positive).

From the above positive samples, a total of 1748 *Salmonella* isolates collected from 20 direct plating samples were processed by the broth breakpoint assay and 55 isolates from 20 direct plating samples were analyzed by the Sensititre^®^ method. Five enrichment isolates were not recoverable when transferred to the 96 well plates containing MH broth. Effects of antimicrobial susceptibility methods on *Salmonella* AMR patterns are reported in Table 2. As compared to the Sensititre^®^ method, the broth breakpoint method reported higher resistance to chloramphenicol (60.3% vs. 25.5%), ciprofloxacin (16.7% vs. 5.5%), sulfisoxazole (61.9% vs. 9.1%), and the combination of trimethoprim and sulfamethoxazole (65.9% vs.12.7%), and lower resistance to azithromycin (15.5% vs. 32.7%), meropenem (16.5% vs. 43.6%), and streptomycin (13.4% vs. 32.7%) in poultry *Salmonella* (all *p* < 0.05).

To reduce the effects of sampling size of isolates on *Salmonella* AMR between AST methods, an equal number of isolates (55 cfu) were randomly selected from the broth breakpoint method and their AMR profiles were compared to those reported by the Sensititre^®^ system (Table 3). Similar results were observed here between the broth breakpoint method and the Sensititre^®^ system, as compared to the comparisons above. As compared to the Sensititre^®^ method, the broth breakpoint method reported higher resistance to chloramphenicol (60.0% vs. 25.5%), ciprofloxacin (27.3% vs. 5.5%), sulfisoxazole (72.7% vs. 9.1%), and the combination of trimethoprim and sulfamethoxazole (61.8% vs. 12.7%), and lower resistance to azithromycin (14.5% vs. 32.7%), meropenem (12.7% vs. 43.6%), and streptomycin (12.7% vs. 32.7%) in poultry *Salmonella* (all *p* < 0.05). Randomized selections and comparisons were conducted five times and similar results were generated (data not shown).

A total of 222 isolates were cultured from 38 samples via RV selective secondary enrichment and assayed for susceptibility in the broth breakpoint assay (108 isolates, Table 4) and the Sensititre^®^ method (114 isolates). The broth breakpoint assay reported a higher prevalence of resistance to the combination of amoxicillin and clavulanic acid (32.1% vs. 5.3%), ampicillin (56.6% vs. 5.3%), cefoxitin (59.4% vs. 5.3%), chloramphenicol (67% vs. 5.3%), meropenem (74.5% vs. 5.3%), sulfisoxazole (98.1% vs. 5.3%), and the combination of trimethoprim and sulfamethoxazole (31.1% vs. 5.3%) and lower prevalence of resistance to streptomycin (75.5% vs. 97.4%, all *p* < 0.050) when compared to the Sensititre^®^ method, respectively.

A nonparametric analysis was conducted to analyze the effects of culturing method on the *Salmonella* AMR pattern generated from the Sensititre^®^ platform (*n* = 14, Table 5). *Salmonella* isolates collected from direct plating demonstrated higher resistance to the combination amoxicillin and clavulanic acid, ampicillin, azithromycin, cefoxitin, ceftriaxone, nalidixic acid, and meropenem, and lower resistance to streptomycin and tetracycline.

In trial two, 96 broiler ceca were screened and generated 48 presumptive positive samples. The 48 presumptive samples were processed for direct plating and RV enrichment. Each sample grew on XLD agar plates with morphologies typical of *Salmonella*, indicating a 50% (48/96) incidence in our convenience samples.

Fifteen samples were randomly selected and assayed for AMR through both AST methods in trial two. To eliminate the effects of *Salmonella* isolate diversity on AST outcomes, each randomly selected *Salmonella* isolate was struck onto a new agar plate and the same colonies were introduced into the breakpoint assay and Sensititre^®^ assay in parallel. Major effects and interactions of the susceptibility test and culturing methodology on *Salmonella* AMR are reported in Table 6. After accounting for isolate diversity between AST methods, neither the interaction nor the culturing method affected *Salmonella* AMR observations. The broth breakpoint assay reported lower resistance to chloramphenicol (0% vs. 2.08%), gentamicin (0% vs. 3.75%), and higher resistance to nalidixic acid (5.0% vs. 0.83%) as compared to the Sensititre^®^ method (all *p* < 0.05).

## 4. Discussion

Poultry products and the poultry production environment are significant sources of *Salmonella* that may be resistant to antibiotics [16,17]. In the current study, *Salmonella* prevalence of the sampled broiler population was 28.6% (59/206) in December 2018 and 50.0% (48/96) in August 2019. The Food Safety and Inspection Service (**FSIS**) tested 575 broiler ceca and reported a 17.9% prevalence of *Salmonella* in 2014 [18]. Furthermore, FSIS tested whole carcasses and parts, *Salmonella* prevalence in whole carcasses, quarters, parts, and comminuted chicken was 3.59%, 8.99%, 8.36%, 31.21%, respectively, in 2019 [19]. *Salmonella* prevalence in carcass rinses has been reported as high as 54.09% [19]. A survey of 15 broiler processing plants indicated that even after sanitization procedures, *Salmonella* prevalence remained at a range of 7.4% to 29.69% [20]. A research study indicated that 7% of poultry origin *Salmonella* isolates in the southeastern United States exhibited resistance to at least one antimicrobial [21]. The FSIS 2014 reported that 13% of *Salmonella* isolates in 2014 were multidrug-resistant [18]. From 2015 to 2017, there was an increase in multidrug-resistant *Salmonella* recovered from the chicken carcass (9.5% to 18%) and chicken cecum samples (15% to 25%) [22].

The NARMS established by the Centers for Disease Control and Prevention, USDA, and FDA, monitors *Salmonella* and other critical enteric bacteria in humans, animals, and meat products for their resistance to various antimicrobials of human and veterinary importance. However, according to the updated NARMS protocol [13], only one single *Salmonella* isolate per human source, retail meat, or food animal-related sample was assayed for AST via automated systems (Sensititre^TM^ and Vitek 2 Compact). Limited isolates per biological sample are tested due to labor, consumables, and general expense, which brings up a concern about how mixed *Salmonella* populations containing clones that dominate during enrichment impact the surveillance program. A serotyping/sequencing method revealed 91% of poultry cecum samples harbored multiple serovars and one single sample could have four different serovars [23]. Thus, it is possible that examining a single isolate from one animal is not enough to determine the range of AMR patterns exhibited by *Salmonella* at premises. This study utilized an alternative method, a high throughput broth breakpoint assay, which allowed up to 94 isolates to be tested per plate with one antimicrobial compound at the CLSI/NARMS resistance breakpoint concentration [14]. By saving preparation time, antibiotics (other CLSI/NARMS recommended concentrations), and related consumables (plates, Mueller-Hinton broth, tips, etc.), the broth breakpoint assay is relatively inexpensive.

The broth breakpoint method reported significantly different AMR pattern determinations from a population of samples as compared to the paired outcomes on the Sensititre^®^ platform in trial one. Increased resistance to chloramphenicol, ciprofloxacin, sulfisoxazole, and the combination of trimethoprim and sulfamethoxazole and decreased resistance to azithromycin, meropenem, and streptomycin were observed in poultry *Salmonella* when using the broth breakpoint method as compared to the Sensititre^®^ method. A potential explanation for the difference observed in AMR patterns may be partially attributed to different isolate sample sizes. Increasing the number of isolates assayed could improve the overall detection of phenotypic AMR diversity in *Salmonella* within and between animal populations. However, exactly how many isolates should be included to improve coverage is unknown. This number will depend on the cost of the AST method used, the feasibility of sample collection, and the AMR diversity of the target bacteria.

When equalizing a sample size (Table 3) or limiting the sampling size to three isolates per RV enrichment sample (Table 4), the broth breakpoint assay still reported increased resistance to chloramphenicol, sulfisoxazole, and the combination of trimethoprim and sulfamethoxazole, and lower resistance to streptomycin as compared to the Sensititre^®^ method. These observed differences in AMR patterns for the population may be attributable to diversity among the *Salmonella* isolates within and between individual ceca samples. Antimicrobial susceptibility testing methods have been largely developed to guide clinicians toward improved individual clinical treatment outcomes and are not generally conducive to high throughput testing for the purposes of surveillance and population research. The observations reported herein may therefore have important implications for understanding the underlying and emerging antimicrobial resistance trends in surveillance programs for agricultural animals.

Unexpected resistance to meropenem was found with both the Sensititre^®^ method and breakpoint assay. Meropenem belongs to carbapenems, a critically important antibiotic class. In the current study, cecum samples were positive by PCR, however, not all *Salmonella* isolates were biochemically and serologically confirmed. A fault of the breakpoint method should be noted that it is not feasible to biochemically confirm every single isolate and that resistance could be inflated due to error processing morphologically similar isolates that were not *Salmonella*. While researchers were careful to pick individual and isolated colonies, it is also possible that mixed cultures may have caused some of these unanticipated results.

After attempting to account for the differences among *Salmonella* isolate diversity and sample size introduced to each assay, the broth breakpoint assay demonstrated similar susceptibility results to the Sensititre^®^ method in trial two except for lower resistance to gentamicin (0.00% vs. 3.75%) and chloramphenicol (0.00% vs. 2.08%) and a higher resistance to nalidixic acid (5.0% vs. 0.84%), respectively. These differences may have been caused by methodological variations of the two AST methods, including antimicrobial compound preparation (dry powder in the Sensititre^®^ plate vs. wet solution in the broth breakpoint plate), inoculum preparation (overnight incubation onto blood agar plates vs. in the 96-well plate with MH broth), reaction volume (50 µL in each well of the Sensititre^®^ plate vs. 200 µL in each well of broth breakpoint ), reaction concentration (1.4 × 10^5^ cfu/mL, estimated by McFarland standards vs. 1.5 × 10^6^ cfu/mL, estimated by OD_600_ reading), and interpretation of AMR results (Vizion^TM^ Digital MIC Viewing System vs. OD600 reading). In early comparative studies, the Sensititre^®^ platform was compared to other susceptibility tests (MicroScan, Vitek2, E-test, and Micronaut strip) for their essential and categorical agreements with broth micro-dilution method [24,25,26,27]. The Sensititre^®^ method exhibited 97.1% essential agreement with the reference broth micro-dilution method in measuring *Klebsiella pneumoniae* resistance to polymyxins and 92.5% for colistin [26], and acceptable categorical agreements (>90%) in measuring colistin and polymyxin B resistance in *Enterobacteriaceae* [24,27]. No major errors were found with the Sensititre^®^ system [25]. Other antimicrobial susceptibility methodologies like the E-test and disk-diffusion were also considered less accurate than the reference agar dilution procedure [28,29]. Further research is warranted to determine the comparative accuracy of the high throughput broth breakpoint assay as compared to agar/broth dilution methods [30] when considering high volumes of diverse isolate testing.

Another important finding in this study was that observed resistance to multiple antimicrobial compounds was associated with culture methods. *Salmonella* detection and isolation bias imparted by microbiological culture methods and media have been documented and potential implications to the accuracy of surveillance programs were discussed [31,32,33,34]. Singer et al. [31] reported cultivation media selected preferentially for specific strains of *Salmonella* in heterogeneous cultures, such as a tetrathionate-RV protocol and a bovine fecal culture. A *Salmonella* Newport strain was more competitive than two *Salmonella* Typhimurium and one *Salmonella* Enteritidis strains in both media [31]. Concentrations of *Salmonella* Enteritidis at the end of the enrichment period were eight-fold higher in tetrathionate as compared to those in RV, whereas *Salmonella* Schwarzengrund and Reading preferentially enriched in RV [32]. Goriski [33] reported the bias of selective enrichment media (RV and RV nutrient-dense version) on the types of *Salmonella* isolates from mixed strain culture and cattle fecal cultures. Different formulations of RV media reported different patterns of strain dominance [33]. Metagenomic analyses indicated that the enrichment procedure reported significantly different taxonomic profiles with high numbers of putative *Salmonella* sequences as compared to the unenriched samples [34]. A serotype sequence assay revealed the RV enrichment reduced the *Salmonella* serovar diversity [32]. However, whether the RV enrichment altered *Salmonella* AMR prevalence by selectively favoring the growth of certain subgroups of *Salmonella* or directly selecting AMR determinants is unclear.

## 5. Conclusions

Our study measured AMR prevalence in broiler cecum collected from a slaughter plant. These data suggest that the breakpoint method allowed more *Salmonella* isolates to be tested resulting in significantly different AMR patterns than the automated Sensititre^®^ platform. We found that the ability to test more isolates was more representative of the diverse microbial population. This research also demonstrates that the selectivity imparted by pre-enrichment and selective secondary enrichment in RV broth may indeed bias the *Salmonella* detected and isolated, and therefore, subsequently bias the AMR patterns determined through susceptibility testing. While enrichment procedures are necessary to detect *Salmonella* and estimate true underlying prevalence in a population, our data suggest that surveillance and determination of AMR patterns for a population could be severely limited due to the bias imparted by the isolation procedure and AST methodology combined.

## Figures and Tables

**Figure 1 microorganisms-09-01319-f001:**
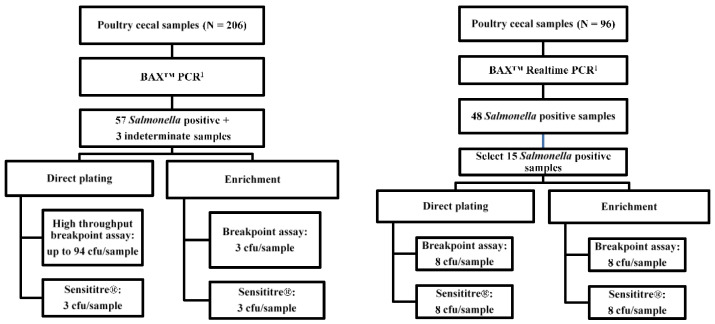
Workflows of trial 1 (**left**) and trial 2 (**right**). ^1^ Cecum samples in trial one were screened for Salmonella presumptive samples using an endpoint assay (BAX^TM^ Salmonella 2 PCR Assay, KIT2012, Hygiena LLC), and cecum samples in trial two were screened via a real-time PCR assay (BAX^TM^ Salmonella Real-time PCR Assay, KIT2006, Hygiena LLC).

**Table 1 microorganisms-09-01319-t001:** Resistant breakpoint concentrations of antimicrobial agents used for *Salmonella*.

Antimicrobial Agent	Resistant Breakpoint (µg/mL)
**Amoxicillin/Clavulanic Acid ^1^**	32/16
**Ampicillin**	32
**Azithromycin**	32
**Cefoxitin**	32
**Ceftriaxone**	4
**Chloramphenicol**	32
**Ciprofloxacin**	1
**Gentamicin**	16
**Meropenem**	4
**Nalidixic Acid**	32
**Streptomycin**	32
**Sulfisoxazole**	512
**Tetracycline**	16
**Trimethoprim/Sulfamethoxazole ^2^**	4/76

Resistant breakpoint concentrations for Salmonella AMR were cited from the National Antimicrobial Resistant Monitoring System under Disease Control and Prevention (https://www.cdc.gov/narms/antibiotics-tested.html). ^1^ the combination of amoxicillin and clavulanic acid was prepared at a 1:2 ratio with 32 µg/mL of amoxicillin and 16 µg/mL of clavulanic acid in the final testing well. ^2^ the combination of trimethoprim and sulfamethoxazole was prepar6ed at 1:19 ratio with 4 µg/mL of trimethoprim and 76 µg/mL of sulfamethoxazole in the final testing well.

**Table 2 microorganisms-09-01319-t002:** Effects of antimicrobial susceptibility testing methodology on resistance ratios of poultry *Salmonella* collected via direct plating in trial one.

Antibiotics	Broth Breakpoint	Sensititre^®^	*X* ^2^	*p* Value
	*n* = 1748	*n* = 55		
**Amoxicillin/Clavulanic acid**	1140 (63.9%)	32 (58.2%)	0.755	0.396
**Ampicillin**	1173 (65.8%)	39 (70.9%)	0.632	0.473
**Azithromycin**	277 (15.5%)	18 (32.7%)	11.72	0.002
**Cefoxitin**	1344 (75.3%)	37 (67.3%)	1.86	0.204
**Ceftriaxone**	1129 (63.3%)	31 (56.4%)	1.09	0.322
**Chloramphenicol**	1075(60.3%)	14 (25.5%)	26.76	<0.001
**Ciprofloxacin**	298 (16.7%)	3 (5.5%)	4.93	0.025
**Gentamicin**	113 (6.33%)	3 (5.5%)	0.070	0.990
**Nalidixic Acid**	477 (26.7%)	12 (21.8%)	0.662	0.535
**Meropenem**	295 (16.5%)	24 (43.6%)	27.33	<0.001
**Streptomycin**	239 (13.4%)	18 (32.7%)	16.58	<0.001
**Sulfisoxazole**	1104 (61.9%)	5 (9.1%)	62.12	<0.001
**Tetracycline**	393 (22.0%)	13 (23.6%)	0.080	0.743
**Trimethoprim/Sulfamethoxazole**	1176 (65.9%)	7 (12.7%)	65.79	<0.001

*Salmonella* isolates were collected from directly plating 20 positive poultry ceca dilutions onto BGA agar plates in trial one. Up to 94 *Salmonella* isolates per cecum sample were tested via the broth breakpoint assay and 1748 isolates were recovered. Three *Salmonella* isolates per cecum sample were randomly selected and tested using the Sensititre^®^ system.

**Table 3 microorganisms-09-01319-t003:** Effects of antimicrobial susceptibility testing methodology on resistance ratios of poultry *Salmonella* collected via direct plating in trial one (equal sample size).

Antibiotics	Broth Breakpoint	Sensititre^®^	*X^2^*	*p* Value
	*n* = 55	*n* = 55		
**Amoxicillin/Clavulanic acid**	33 (60.0%)	32 (58.2%)	0.038	0.999
**Ampicillin**	32 (58.2%)	39 (70.9%)	1.96	0.232
**Azithromycin**	8 (14.5%)	18 (32.7%)	5.04	0.043
**Cefoxitin**	41 (74.5%)	37 (67.3%)	0.705	0.529
**Ceftriaxone**	33 (60.0%)	31 (56.4%)	0.149	0.847
**Chloramphenicol**	33 (60.0%)	14 (25.5%)	13.41	<0.001
**Ciprofloxacin**	15 (27.3%)	3 (5.5%)	9.56	0.004
**Gentamicin**	4 (7.3%)	3 (5.5%)	0.153	0.999
**Nalidixic Acid**	14 (25.5%)	12 (21.8%)	0.202	0.823
**Meropenem**	7 (12.7%)	24 (43.6%)	12.98	<0.001
**Streptomycin**	7 (12.7%)	18 (32.7%)	6.26	0.022
**Sulfisoxazole**	40 (72.7%)	5 (9.1%)	46.06	<0.001
**Tetracycline**	16 (29.1%)	13 (23.6%)	0.422	0.666
**Trimethoprim/Sulfamethoxazole**	34 (61.8%)	7 (12.7%)	28.35	<0.001

*Salmonella* isolates were collected from directly plating 20 positive poultry ceca dilutions onto BGA agar plates in trial one. Up to 94 *Salmonella* isolates per cecum sample were tested via the broth breakpoint assay and 55 isolates were randomly selected for comparison. Three *Salmonella* isolates per cecum sample were randomly selected and tested using the Sensititre^®^ system. Randomization was conducted five times and similar comparison results were generated.

**Table 4 microorganisms-09-01319-t004:** Effects of antimicrobial susceptibility testing methodology on resistance ratios of poultry *Salmonella* isolated via selective enrichment in trial one.

Antibiotics	Broth Breakpoint	Sensititre^®^		
	*n* = 108	*n* = 114	*X* ^2^	*p* Value
**Amoxicillin/Clavulanic acid**	34 (32.1%)	6 (5.3%)	26.54	<0.001
**Ampicillin**	60 (56.6%)	6 (5.3%)	68.94	<0.001
**Azithromycin**	13 (12.3%)	6 (5.3%)	3.41	0.091
**Cefoxitin**	63 (59.4%)	6 (5.3%)	74.88	<0.001
**Ceftriaxone**	5 (4.7%)	9 (7.9%)	0.93	0.413
**Chloramphenicol**	71 (67.0%)	6 (5.3%)	91.97	<0.001
**Ciprofloxacin**	4 (3.8%)	0 (0%)	NA	NA
**Gentamicin**	0 (0%)	0 (0%)	NA	NA
**Nalidixic Acid**	6 (5.7%)	0 (0%)	NA	NA
**Meropenem**	79 (74.5%)	6 (5.3%)	111.15	<0.001
**Streptomycin**	80 (75.5%)	111 (97.4%)	23.01	<0.001
**Sulfisoxazole**	104 (98.1%)	6 (5.3%)	189.41	<0.001
**Tetracycline**	106 (100%)	109 (95.6%)	NA	NA
**Trimethoprim/Sulfamethoxazole**	33 (31.1%)	6 (5.3%)	25.20	<0.001

NA: not available. When 50% of the cells have expected counts less than five, Chi-square is not a valid test. *Salmonella* isolates were collected from 38 poultry ceca enrichments (RV) in trial one. Three *Salmonella* isolates per cecum sample were tested via the broth breakpoint assay and another three isolates via the Sensititre^®^ method. Six colonies, transferred from the enrichment plates, did not grow when passaged to the Mueller-Hinton containing 96-well plates and only 108 colonies were tested for AMR via broth breakpoint assay.

**Table 5 microorganisms-09-01319-t005:** Effects of culturing method on poultry *Salmonella* antimicrobial resistance in trial one.

	Wilcoxon Score		
Antibiotics	Direct Plating	Enrichment	*p* Value
**Amoxicillin/Clavulanic acid**	18.93	10.07	0.001
**Ampicillin**	18.96	10.04	0.001
**Azithromycin**	17.36	11.64	0.023
**Cefoxitin**	18.96	10.04	0.001
**Ceftriaxone**	18.89	10.11	0.001
**Chloramphenicol**	16.39	12.61	0.096
**Ciprofloxacin**	15.00	14.00	0.353
**Gentamicin**	15.50	13.50	0.165
**Nalidixic Acid**	17.00	12.00	0.017
**Meropenem**	17.86	11.14	0.010
**Streptomycin**	9.29	19.71	<.001
**Sulfisoxazole**	14.96	14.04	0.608
**Tetracycline**	8.61	20.39	<.001
**Trimethoprim/Sulfamethoxazole**	15.89	13.11	0.191

Resistance colony counts through the Sensititre^®^ assay from direct plating and those counts from RV enrichment were paired through the Wilcoxon ranking test (*n* = 14 birds). Mean Wilcoxon scores and *p*-values were presented.

**Table 6 microorganisms-09-01319-t006:** Effects of antimicrobial resistance methodology and bacteria culturing method on *Salmonella* antimicrobial resistance patterns in trial two.

								*p* Value	
Methodology	Sensititre^®^	Broth Breakpoint	SEM	Direct Plating	Enrichment	SEM	Method	Culture	Method × Culture
**Amoxicillin/Clavulanic acid**	12.1%	10.0%	3.65%	10.8%	11.3%	3.65%	0.688	0.936	0.574
**Ampicillin**	7.9%	2.9%	2.16%	5.8%	5.0%	2.16%	0.107	0.786	0.587
**Azithromycin**	35.8%	37.9%	7.79%	43.3%	30.4%	7.79%	0.851	0.246	0.910
**Ceftriaxone**	5.4%	0.8%	1.96%	2.9%	3.3%	1.96%	0.103	0.881	0.881
**Cefoxitin**	0.4%	0.0%	0.30%	0.0%	0.4%	0.30%	0.322	0.322	0.322
**Chloramphenicol**	2.08% ^a^	0.00% ^b^	0.62%	0.8%	1.3%	0.62%	0.021	0.637	0.637
**Gentamicin**	3.75% ^a^	0.00% ^b^	1.03%	2.9%	0.8%	1.03%	0.013	0.159	0.159
**Meropenem**	2.5%	4.2%	1.73%	1.7%	5.0%	1.73%	0.498	0.178	0.498
**Nalidixic acid**	0.83% ^b^	5.00% ^a^	1.31%	3.3%	2.5%	1.31%	0.028	0.654	0.182
**Streptomycin**	72.1%	75.0%	5.43%	70.4%	76.7%	5.43%	0.705	0.419	0.705
**Trimethoprim/Sulphamethoxazole**	4.6%	4.6%	3.35%	8.3%	0.8%	3.35%	0.999	0.119	0.990

^a,b^ different subscripts in a row stand for a significant difference at α = 0.05. Means from interactions are not listed due to a lack of significant difference. A total of 15 *Salmonella* positive broiler cecal samples were selected in trial two. For each cecum sample, eight isolates from direct plating and another eight from enrichment were tested by the Sensititre^®^ method and the broth breakpoint method (four treatments = two culture procedures × two AST methodologies, *n* = 15 birds/treatment). In trial 2, all *Salmonella* colonies were resistant to tetracycline (100%) and most *Salmonella* colonies (>99%) across treatments were susceptible to ciprofloxacin and sulfisoxazole. Results were not generated since insufficient variation in those data to perform ANOVA analysis.

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
