# Peer review of "Determination of Antimicrobial Resistance Patterns in Salmonella from Commercial Poultry as Influenced by Microbiological Culture and Antimicrobial Susceptibility Testing Methods"

_microorganisms, 2021, doi:10.3390/microorganisms9061319_

Round 1

Reviewer 1 Report

The study is an interesting comparison of 2 methods employed for the detection of resistant Salmonella strains. The manuscript shows that the resistance patterns provided by both tests are not exactly the same for several antimicrobials, may cast doubt on the results obtained with the method evaluated.

I have several comments on information shown in the study:

  1. Table 2. It is not clear to me that a sample with n = 1748 isolates and another one with only 55 are statistically comparable.
  2. The authors state as a hypothesis in the objective of the manuscript that the broth breakpoint method is more representative when assessing resistance profiles. However, the hypothesis is not reflected in the discussion and conclusion of the manuscript. It is more representative/accurate or not? Or otherwise, given the differences observed in the results obtained with both tests, is the Sensititre system comparable to the broth breakpoint method?
  3. If for other microorganisms the Sensititre system has yielded results quite similar to other methods with which it has been compared, could the differences obtained in this paper be related to the microorganism tested or the procedure used in sample preparation?
  4. Table 3 should appear in the text after being described in the text.
  5. There is an unnumbered reference between reference 17 and 18. Therefore, all references from number 17 onwards should be renumbered.

Author Response

Dear Reviewer,

Thanks you for your efforts to improve this publication.  Please find our rebuttal to your comments attached.

Sincerely,

Morgan Farnell

Reviewer 2 Report

I evaluate this article as very interesting with a very strong impact on the laboratory practice and the way of results evaluation. I have only two recommandations for deeping the discussion and I would like authors will add one item of additional information.

I appreciate highly this work and its impact on everyday´s laboratory practice.

But I would like to ask the authors to make the discussion more deeper and to answer these two followig questions and to add one item of information. 

The first question : The incoherence of the results of broth breakpoint assay and Sensititre is explained deeply in the connection to the choice of tested isolates and the way of their isolation (using direct plating or selective enrichment etc.). But could you discuss more also the differences between these two AST methods ?  They differ mainly in the procedure for the inoculum preparation (overnight grown culture for broth breakpoint assay, while 0.5 McFarland diluted four times in case of Sensititre). What are your ideas about the impact of these different methods for the inoculum preparation on the ability of bacterial isolates to grow in the presence of the same antibiotic concentration (which could influence the evaluation of their antibiotic sensitivity) ? 

At second: What is your general requirement for the practice ? Do you suggest some improvements or modifications in tested AST methods in order to increase the reliability of such testing ? As example to how many items to increase the amount of tested isolates to increase reliability of these AST  for some antibiotics or to improve some methods in some steps ?

One item of information: In lines 365-368 you mention that AST methods have been developed to guide clinicians. Is there some source of information for the distribution of MICs in non-clinical bacterial strains ? If yes, could you mention it and involve in your discussion ?

Author Response

Dear Reviewer,

Thank you for your efforts to improve our publication.  Please see our reply in the attached rebuttal letter.

Sincerely,

Morgan Farnell
